# Visible Light Communication: An Investigation of LED Non-Linearity Effects on VLC Utilising C-OFDM

**Jummah Abdulwali \*** 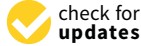 **and Said Boussakta**

Communication and Signal Processing at the School of Engineering, Newcastle University, Newcastle NE1 7RU, UK; said.boussakta@newcastle.ac.uk
* Correspondence: j.a.m.abdulwali2@newcastle.ac.uk

**Abstract:** The electro-optic output of light-emitting diodes commonly used in visible light communication systems is generally nonlinear in nature. It is particularly problematic when using advanced modulation formats, such as orthogonal frequency-division multiplexing (OFDM), which have a high peak-to-average power ratio due to clipping and distortion. In this work, we introduce the so-called C-transform to the system architecture, which utilises a Walsh–Hadamard matrix in conjunction with a discrete cosine transform to deterministically spread the information and reduce the peak-to-average power ratio (PAPR). Several bias points along the electro-optic transfer function were selected for comparison purposes, and the new transform was compared with more traditional formulations of OFDM. This paper determines that the C-transform-based OFDM demonstrated the highest degree of independence from the non-linearity and yielded superior bit-error rate (BER) results. We note an improvement of ~2.5 dB in the power penalty at a BER of $10^{-4}$ in comparison to OFDM.

**Keywords:** bias point; non-linearity; orthogonal frequency division multiplexing (OFDM); peak-to-average power ratio (PAPR); visible light communication (VLC)

## 1. Introduction

Orthogonal frequency division multiplexing (OFDM) is a popular candidate technology for visible light communication (VLC) systems due to its robustness against multipath propagation and its high spectral efficiency. For VLC systems, there are two types of OFDM: asymmetrically clipped optical OFDM (ACO-OFDM) [1] and DC-biased optical OFDM (DCO-OFDM) [2]. The transmitted signal is turned positive in ACO-OFDM by zero-clipping the original bipolar OFDM signal and broadcasting only the positive components. In DCO-OFDM, the signal is made positive by adding a DC bias. Only the odd subcarriers convey data symbols in ACO-OFDM, while all subcarriers send data symbols in DCO-OFDM. DCO-OFDM is less efficient than ACO-OFDM in terms of the average optical power. In contrast, ACO-OFDM utilises only half of the subcarriers for data transmission, making DCO-OFDM more inefficient in terms of bandwidth [3]. However, the light-emitting diodes (LEDs) used in VLC have two key challenges; firstly, the intrinsic low-pass nature of LEDs leads to bandwidth limitations. It has traditionally been the fundamental challenge associated with VLC [4,5].

This challenge has mainly been overcome with advanced modulation formats such as OFDM, delivering a higher spectral efficiency [6,7]. Digital signal processing algorithms, including equalisers and artificial neural network-based classifiers, have also pushed data rates into the Gb/s regions [8,9]. The second major LED-based challenge relates to the non-linearity in the electro-optic transfer function (see Figure 1), which causes an issue because the signals are intensity-modulated onto the optical power [10]. Generally, OFDM systems have a high peak-to-average power ratio (PAPR) due to the sum of many statistically independent sinusoids, obviously causing issues when dealing with digital-to-analogue converters and nonlinear transfer functions. Therefore, the received signal suffers from significant distortions in incorrectly designed OFDM-based VLC systems [10,11].

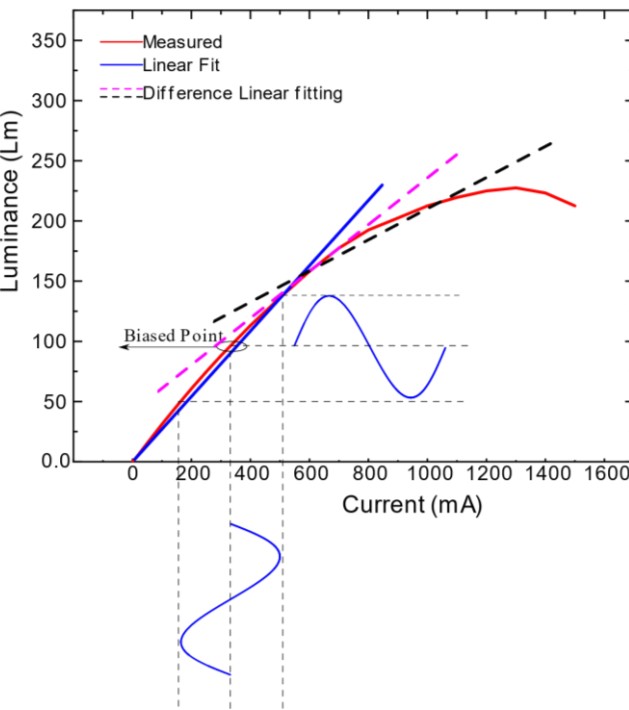

**Figure 1.** Nonlinear LED transfer function.

In general, two methods can be employed to compensate for LED non-linearities. The first choice is to use a PAPR reduction process as proposed in [12], which operates by applying a semidefinite relaxation approach to tone injection. The second approach is to linearise the LED nonlinear transfer characteristics [13]. Clipping, coding, selected mapping (SLM), partial transmit sequences (PTS), tone reservation (TR), tone injection (TI), as well as other techniques are applied to solve the PAPR issue in conventional radio frequency (RF) communication systems, as summarised in [14,15]. In the RF domain, power amplifiers introduce a nonlinear distortion. Furthermore, DC-biased OFDM requires a high bias to avoid low-end clipping, which significantly degrades the system's power efficiency.

Consequently, reducing the PAPR in the VLC OFDM technique is even more important. In asymmetrically clipped OFDM, subcarrier mapping is utilised to guarantee positive OFDM signals. Therefore, traditional PAPR reduction approaches may not be suitable [12]. A pilot-assisted PAPR reduction technique has been proposed in [16], which can achieve a better PAPR efficiency compared to SLM. However, depending on the density of pilot symbols, the pilot-assisted approach causes some data rate loss. A significant PAPR reduction achieved via an exponential nonlinear commanding algorithm was proposed in [17]. This method achieved a good PAPR reduction, but led to a poor bit-error rate (BER).

There are other OFDM architectures that have become the focus of attention recently, and the most important of these is fast OFDM. The time-domain signal of fast OFDM is generated utilising an inverse discrete cosine transform (IDCT), rather than the inverse fast Fourier transform (IFFT) that is used in OFDM. It has significant benefits for VLC, in particular due to forsaking complex modulation formats in favour of actual ones such as pulse-amplitude modulation (PAM), allowing subcarrier spacing to be reduced [18,19].

In this paper, we propose C-transform-based OFDM. This architecture includes combining two transforms. The time-domain signal stream is obtained as an IDCT preceded by the Walsh–Hadamard transform [20].

In this paper, a comparative study is conducted to examine the effect of LED non-linearities. Furthermore, LEDs have a minimum threshold voltage, referred to as the turn-on voltage (TOV), above which current flow and light emission begin. As a result, the LED is in a cut-off area below the TOV and does not conduct a current. Above the TOV, however, current flow and light output rapidly increase with voltage. The LED produces light linearly with the

driving current (see Figure 2, the measured L–I–V curves of the LED). As a result, the DC and AC currents must be regulated appropriately to prevent deterioration in output illumination or complete failure in the worst-case scenario. Thus, different OFDM schemes were tested, namely, conventional DC-biased OFDM, F-OFDM and C-OFDM. To compare these systems, we varied the DC bias and peak-to-peak AC signal amplitude in order to understand the impact of the LED electro-optic non-linearity on the BER performance of the optical OFDM architectures, whilst considering the PAPR. We find that the C-OFDM system outperforms. The proposed system has the advantages of being resistant to LED non-linearity dispersion and having a lower PAPR than other systems.

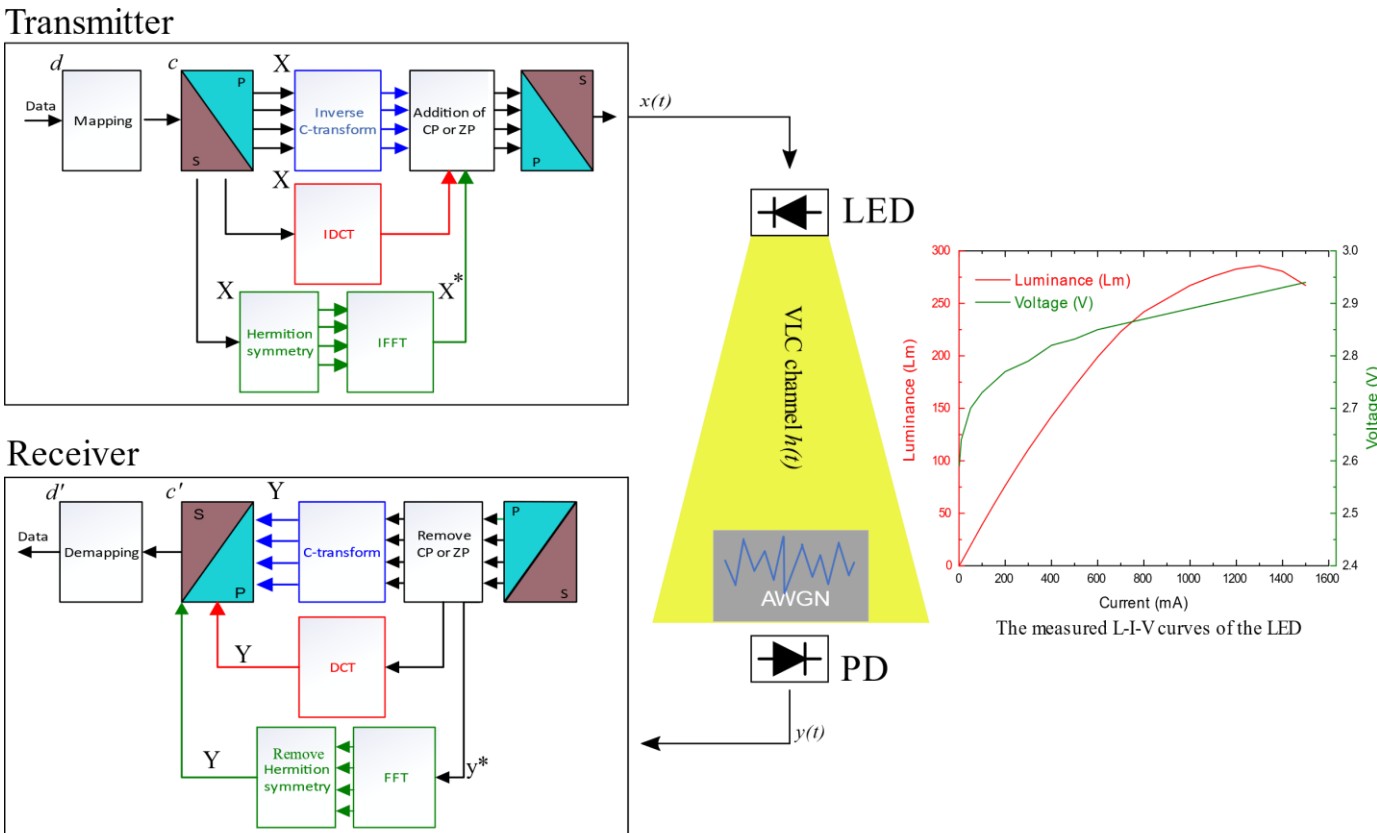

**Figure 2.** System block diagram.

The remainder of the paper is organised as follows: In Section 2, the test setup is described. The results are discussed in Section 3, and the conclusion is presented in Section 4.

## 2. Test Setup

On the transmitter side, a block diagram for each of the OFDM systems tested is shown in Figure 2. Firstly, the architectures were very similar and the C-OFDM (blue) and F-OFDM (red) variations were slightly different from the conventional OFDM approach (green). An independent data stream (d) was mapped onto the modulation alphabet for all systems, which was selected specifically for the format. For the F-OFDM and C-OFDM systems, a real-valued modulation format, such as a pulse amplitude modulation (PAM), had to be considered because the cosine transform accepted a real input only and produced a real output. For the conventional OFDM system, quadrature amplitude modulation (QAM) could be utilised. The number of bits-per-symbol was, therefore, set to kPAM = kQAM/2 to preserve the equivalent bit rate. Moreover, the DCT transform was spacing the subcarriers by df = 1/2T [6,19,21], although in IFFT, the spacing between the subcarriers was (1/T). Furthermore, all the subcarriers N carried the information using PAM in terms of C-OFDM

and F-OFDM. However, in OFDM, N/2 subcarriers carried information, and N/2 was conjugated to obtain the real output of IFFT using QAM. It is worth mentioning that a fair comparison was performed with F-OFDM; the OFDM was added as a reference.

For each scheme, the data were subsequently passed through a serial-to-parallel converter (*c*) prior to the first scheme, the inverse C-transform in the case of C-OFDM, or in the second one IDCT was applied in the case of F-OFDM (X). Finally, Hermitian symmetry (HS) generation and transformation through IFFT in the case of OFDM formulated the output of the IFFT (X*), a real time-domain signal. HS was considering an additional stage compared to C-OFDM and F-OFDM. The HS concept was defined as follows:

$$S_m \begin{cases} 0 & m = 0 \\ X_m & 0 < m < N/2 \\ 0 & m = N/2 \\ X_m^* & N/2 < m < N-1 \end{cases}. \tag{1}$$

where the $S_m$ is the HS output, $X_m$ is the *m*th parallel data stream and *N* is the number of subcarriers (see Figure 3), then zero-padding (ZP) was proposed as an interesting alternative to cyclic prefix (CP), as presented in [22,23].

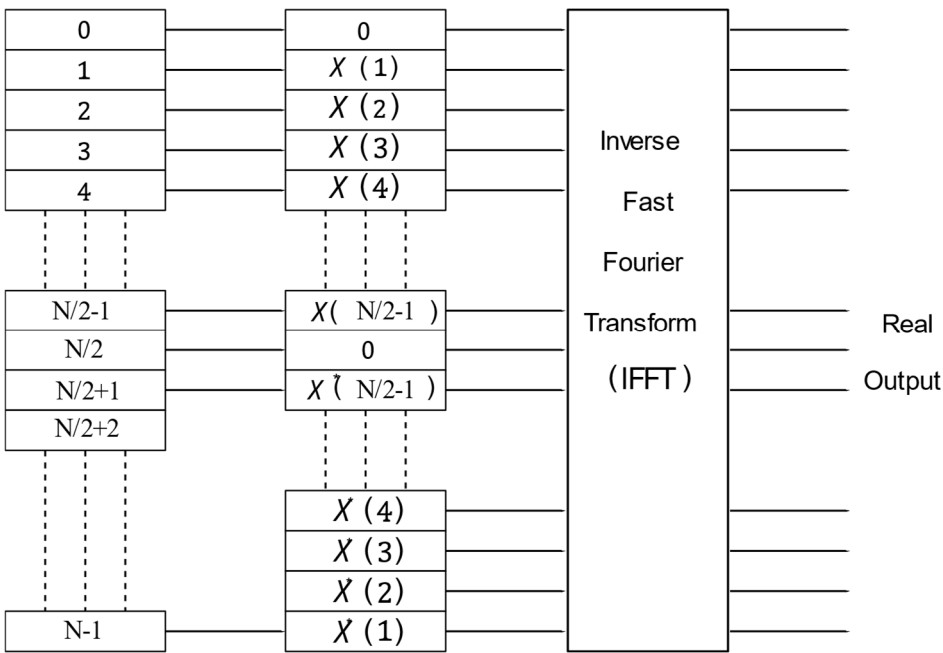

**Figure 3.** Hermitian symmetry (HS).

The proposed system C-OFDM was utilising the C-transform instead of using FFT. The C-transform is a unitary transform which combines the effect of these two transforms, namely, WHT and DCT.

### 2.1. Walsh–Hadamard Transform

The WHT decomposes a signal into a series of orthogonal, rectangular waveforms known as Walsh functions using a real, nonsinusoidal transformation. Rearranging the rows of a Hadamard matrix into a sequence order yields the Walsh–Hadamard matrix. By rearranging the rows of an order eight Hadamard matrix, an order eight Walsh matrix can be created as follows:

$$
H_{8 \times 8} = \begin{bmatrix}
1 & 1 & 1 & 1 & 1 & 1 & 1 & 1 \\
1 & -1 & 1 & -1 & 1 & -1 & 1 & -1 \\
1 & 1 & -1 & -1 & 1 & 1 & -1 & -1 \\
1 & -1 & -1 & 1 & 1 & -1 & -1 & 1 \\
1 & 1 & 1 & 1 & -1 & -1 & -1 & -1 \\
1 & -1 & 1 & -1 & -1 & 1 & -1 & 1 \\
1 & 1 & -1 & -1 & -1 & -1 & 1 & 1 \\
1 & -1 & -1 & 1 & -1 & 1 & 1 & -1
\end{bmatrix}
\begin{matrix}
S_{ch}(0) \\ S_{ch}(7) \\ S_{ch}(3) \\ S_{ch}(4) \\ S_{ch}(1) \\ S_{ch}(6) \\ S_{ch}(2) \\ S_{ch}(5)
\end{matrix}
$$

$$
W_{8 \times 8} = \begin{bmatrix}
1 & 1 & 1 & 1 & 1 & 1 & 1 & 1 \\
1 & 1 & 1 & 1 & -1 & -1 & -1 & -1 \\
1 & 1 & -1 & -1 & -1 & -1 & 1 & 1 \\
1 & 1 & -1 & -1 & 1 & 1 & -1 & -1 \\
1 & -1 & -1 & 1 & 1 & -1 & -1 & 1 \\
1 & -1 & -1 & 1 & -1 & 1 & 1 & -1 \\
1 & -1 & 1 & -1 & -1 & 1 & -1 & 1 \\
1 & -1 & 1 & -1 & 1 & -1 & 1 & -1
\end{bmatrix}
\begin{matrix}
S_{ch}(0) \\ S_{ch}(1) \\ S_{ch}(2) \\ S_{ch}(3) \\ S_{ch}(4) \\ S_{ch}(5) \\ S_{ch}(6) \\ S_{ch}(7)
\end{matrix}
$$

where H is a Hadamard matrix and W is a Walsh matrix and $S_{ch}(7)$ illustrates that the sign was changed seven times in this row. The first element in the last row was 1 and then $-1$, which was the first change; then, $-1$ changed to 1 as the second change in the sign; then, 1 changed to $-1$, which was the third change; this changed seven times. Additionally, $S_{ch}(0)$ designated the signs of the matrix element and remained identical.

Moreover, the WHT matrix's elements could be written as:

$$
W_{x,u} = (-1)^{\sum_{i}^{n-1} x_i u_i}. \tag{2}
$$

where $n = \log_2(N)$, where $x$ and $u$ are the rows and columns positions in the WHT matrix and $x_i * u_i$ is the *i*th product operation of bit-by-bit for the binary number which represents the integer numbers $x$, $u$ and $n$; $-1$ is the number of binary digits in each index. To clarify Equation (2), let us assume that W(6,5) where $x = 6$ and $u = 5$ where $x$ is the sixth row and $u$ is the 5th column in $W_{8 \times 8}$, are presented in binary as 110 and 101, respectively. Appling these numbers in Equation (2) would result in:

$$
W_{x,u} = (-1)^{g(x_i u_i)}
$$

$$
g(6,5) = \sum_{j=o}^{3-1} 6_j * 5_j
$$

$$
g(6,5) = b_0(6) * b_0(5) + b_1(6) * b_1(5) + b_2(6) * b_2(5)
$$

$$
g(6,5) = 0 * 1 + 1 * 0 + 1 * 1 = 1
$$

$$
W(6,5) = (-1)^1 = -1
$$

### 2.2. Discrete Cosine Transform

The discrete cosine transform is a function that expresses a finite sequence of data points as a sum of cosine functions oscillating at various frequencies. DCTs are a type of Fourier-related transform that only utilise a real part. Furthermore, since DCT is a real transform when data mapping is real, using it with the OFDM technique would avoid the in-phase/quadrature-phase (IQ) imbalance. Likewise, the DCT is less affected by the carrier frequency offset than the FFT [24,25]. The DCT matrix could be given as:

$$
\text{DCT}_{k,n} = \sqrt{\frac{2}{N}} \sum_{n=0}^{N-1} \psi_n \cos\left[\frac{\pi(2k+1)n}{2N}\right]. \tag{3}
$$

where $k$ and $n$ are the rows and columns position in the DCT matrix and $\psi_n$ is the normalisation factor, which is determined by:

$$\psi_n = \begin{cases} \frac{1}{\sqrt{2}}, & n = 0 \\ 1, & 0 < n < N - 1 \end{cases}. \tag{4}$$

Therefore, a DCT matrix of size eight could be as follows:

$$\text{DCT}_{8 \times 8} = \begin{bmatrix} 0.35 & 0.35 & 0.35 & 0.35 & 0.35 & 0.35 & 0.35 & 0.35 \\ 0.5 & 0.41 & 0.28 & 0.1 & -0.1 & -0.28 & -0.42 & -0.5 \\ 0.46 & 0.19 & -0.19 & -0.46 & -0.46 & -0.19 & 0.19 & 0.46 \\ 0.41 & -0.1 & -0.5 & -0.28 & 0.28 & 0.5 & 0.1 & -0.42 \\ 0.35 & -0.35 & -0.35 & 0.35 & 0.35 & -0.35 & -0.35 & 0.35 \\ 0.28 & -0.5 & 0.1 & 0.42 & -0.41 & -0.1 & 0.5 & -0.28 \\ 0.19 & -0.46 & 0.46 & -0.19 & -0.19 & 0.46 & -0.46 & 0.19 \\ 0.1 & -0.28 & 0.42 & -0.5 & 0.5 & -0.41 & 0.28 & -0.1 \end{bmatrix}$$

*2.3. C-Transform*

We retrieved the OFDM system that applies DCT and WHT separately from [26]. Moreover, in terms of reducing complexity, both transforms were compacted into one transform: the C-transform was proposed in [27] as follows:

$$C_{8 \times 8} = \left[ \left( \hat{\text{DCT}_{8 \times 8}} \right) \right] \left[ \left\{ \left( \hat{\text{WHT}_{8 \times 8}} \right) \right\}^T \right]$$

where $\hat{(.)}$ denotes the bit-reversed order and $(.)^T$ signifies the transpose. For clarity and without a loss of generality, matrix $8 \times 8$ of the C-transform is shown as follows:

$$C_{8 \times 8} = \begin{bmatrix} 1 & 0 & 0 & 0 & 0 & 0 & 0 & 0 \\ 0 & 1 & 0 & 0 & 0 & 0 & 0 & 0 \\ 0 & 0 & 0.92 & 0.38 & 0 & 0 & 0 & 0 \\ 0 & 0 & -0.38 & 0.92 & 0 & 0 & 0 & 0 \\ 0 & 0 & 0 & 0 & 0.91 & -0.07 & 0.37 & 0.18 \\ 0 & 0 & 0 & 0 & 0.21 & 0.77 & -0.51 & 0.32 \\ 0 & 0 & 0 & 0 & -0.32 & 0.51 & 0.77 & 0.21 \\ 0 & 0 & 0 & 0 & -0.18 & -0.37 & -0.07 & 0.9 - 01 \end{bmatrix}$$

Given that CT is a real orthonormal matrix, its inverse was the same as its transpose [27].

$$\text{ICT} = \hat{\text{CT}}$$

It was evident that CT had a diagonal block structure (DBS). Similarly, more than 65% of its elements were zero, which led to reduced numbers of real additions and multiplications. Therefore, that made it advantageous in terms of reducing the PAPR, as well as the N-class matrix.

On the receiver side, prior to serial-to-parallel conversion and zero padding elimination, AWGN was added to the received data $y(t)$ at the receiver. These data were subsequently injected into the forward C transform (C-OFDM), DCT (F-OFDM) or FFT (conventional OFDM), respectively. The elimination of the Hermitian symmetry was an extra step in relation to FFT-OFDM. Finally, the modulation alphabet was de-mapped from the results.

In this work, we selected several bias points along with the LED electro-optic transfer function and varied the AC level chosen. By running the LED in a quasi-linear segment for its characteristic around the chosen bias point and AC level, the BERs of the F-OFDM and C-OFDM signal were monitored to examine the distortion levels. Consequently, this resulted in a better understanding of the optimal bias conditions.

## 3. Simulation Results and Discussion

The performance of the optical OFDM system was significantly impacted by LED non-linearity. The DC bias point was one of the key characteristics that the distortion it led to depended on. In this section, the results were considered in terms of BER performance and power penalty. The comparison that was evident between C-OFDM and fast OFDM was discussed. It is worth mentioning that the real comparison was between the C-OFDM and F-OFDM, as both used real transforms. However, the comparison of the conventional OFDM provided greater clarification.

### 3.1. BER Performance

Figure 4 demonstrates the BER performance of the conventional 4-QAM OFDM under the operation condition of the 200 AC level and different bias points. The solid black curve represents the theoretical curve BER performance in all coming figures. From Figure 4, it could be observed that the BER managed to meet the designated BER target $10^{-4}$ at $E_b/N_o \sim 20$ dB when the bias point was 100 mA. Moreover, Figure 5 illustrates the BER performance of 2-PAM F-OFDM. It showed a slight improvement and met the BER target at $E_b/N_o < 10$ dB. However, Figure 5 infers that the BER performance when the DC bias point was more the 500 mA failed to meet the BER target $10^{-4}$. On the other hand, Figure 6 illustrates the BER performance of 2-PAM C-OFDM under the same operating condition of the DC bias point. An improvement in the BER was shown in Figure 6 using C-OFDM. An improvement in the BER could be observed in contrast to OFDM, F-OFDM and C-OFDM in Figures 4–6, respectively. The 16-QAM OFDM BER performance was depicted in Figure 7. It showed that the BER target had not been achieved at the low $E_b/N_o$. Figure 8 illustrates the BER performance as a function of $E_b/N_o$ for 4-PAM F-OFDM. In Figure 8, it can be noted that for most of the different bias point values, the BER target was not achieved. The BER performance of 4-PAM C-OFDM was presented in Figure 9. It showed a significant improvement in the BER performance, where all the BER curves managed to meet the BER target at the lowest $E_b/N_o$ in contrast to Figures 7 and 8.

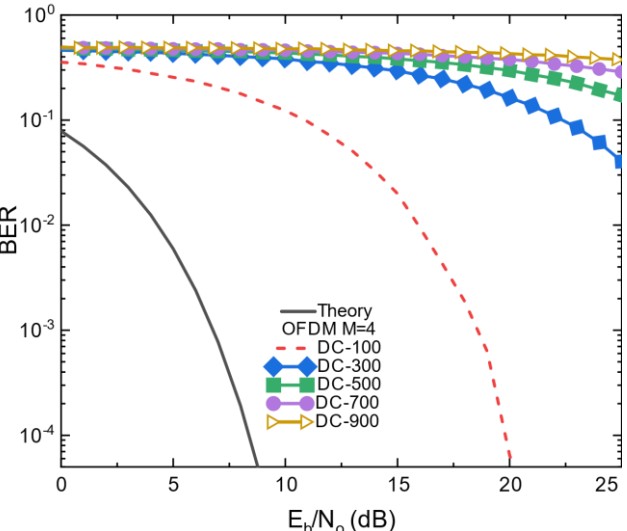

**Figure 4.** BER performance as a function of $E_b/N_o$ for OFDM with 4-QAM and 200 AC.

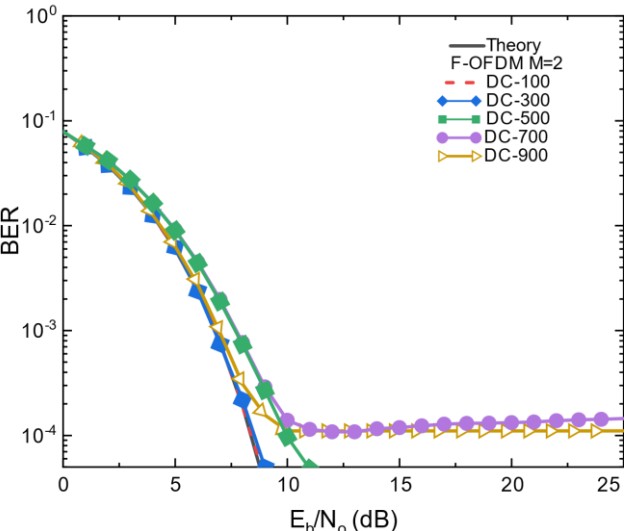

**Figure 5.** BER performance as a function of $E_b/N_o$ for F-OFDM with 2-PAM and 200 AC.

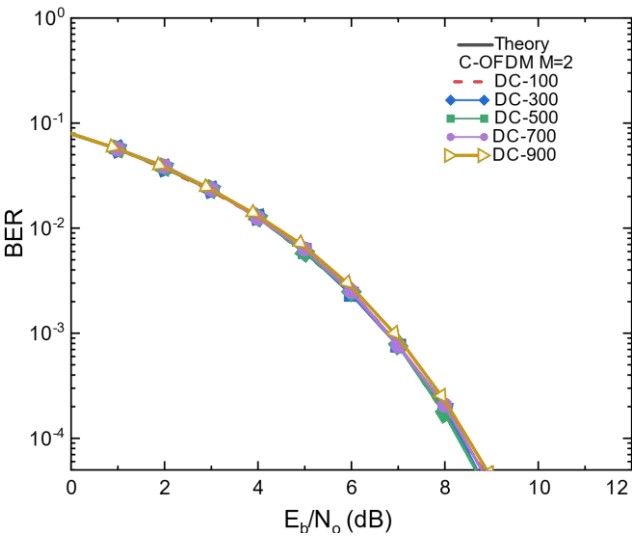

**Figure 6.** BER performance as a function of $E_b/N_o$ for C-OFDM with 2-PAM and 200 AC.

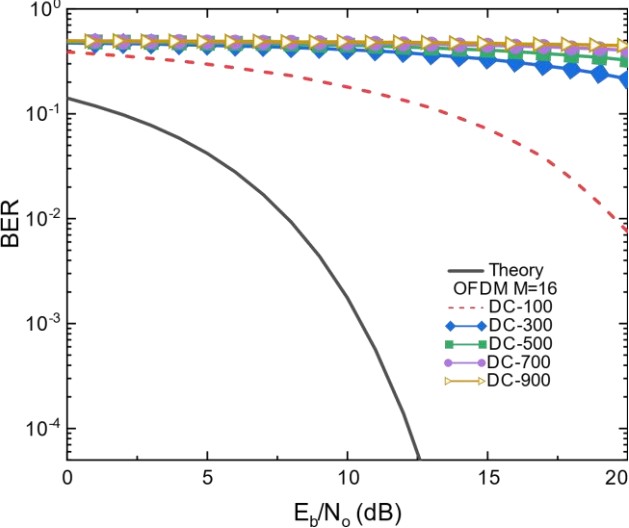

**Figure 7.** BER performance as a function of $E_b/N_o$ for OFDM with 16-QAM and 200 AC.

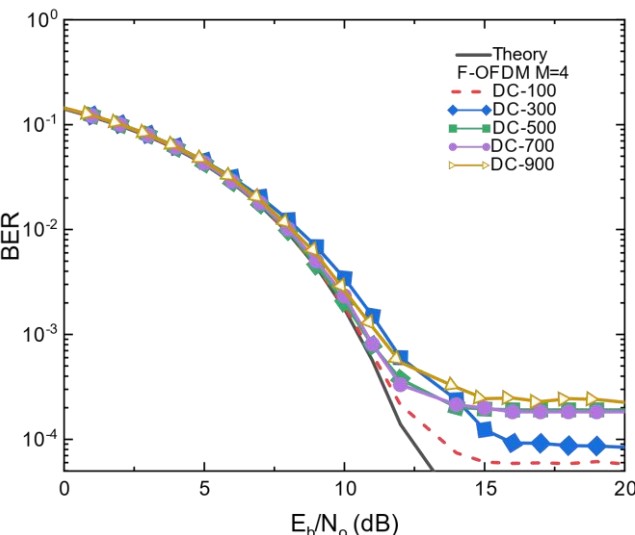

**Figure 8.** BER performance as a function of $E_b/N_o$ for F-OFDM with 4-PAM and 200 AC.

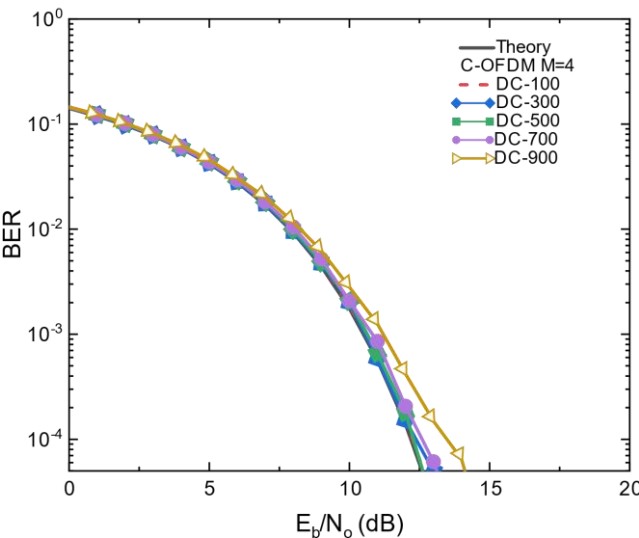

**Figure 9.** BER performance as a function of $E_b/N_o$ for C-OFDM with 4-PAM and 200 AC.

Moreover, for all the bias points, the C-OFDM's BER's performance was better than the conventional OFDM and F-OFDM. Furthermore, Figures 6 and 9 show the BER performance of the proposed systems with 2-PAM and 4-PAM, respectively, where both showed an improved BER performance in contrast to the typical OFDM and F-OFDM. Finally, we changed the AC level to 600 mA, as seen in Figures 10–12, for OFDM, F-OFDM and C-OFDM, respectively. It can be seen from Figures 10 and 12 that an ~10 dB improvement was achieved at the first bias point compared to conventional OFDM and C-OFDM. At DC bias 300 and 500, an improvement of ~4 dB and 9 dB was achieved when compared between C-OFDM and F-OFDM, respectively, in Figures 11 and 12. However, the BER did not reach the target using F-OFDM with the 600 DC level and an ac bias point greater than 500 mA.

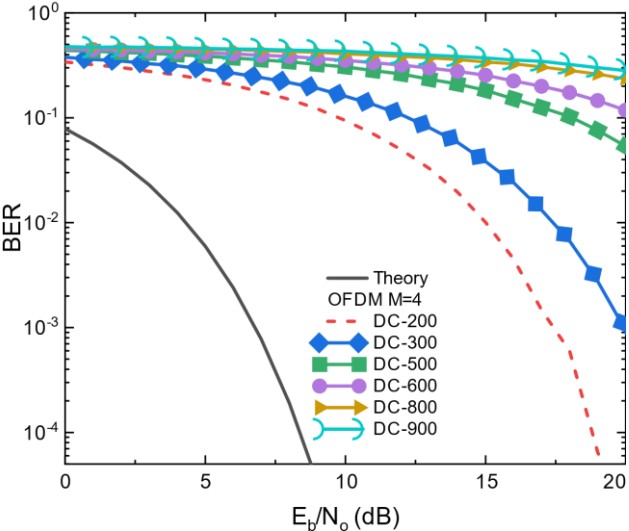

**Figure 10.** BER performance as a function of $E_b/N_o$ for OFDM with 4-QAM and 600 AC.

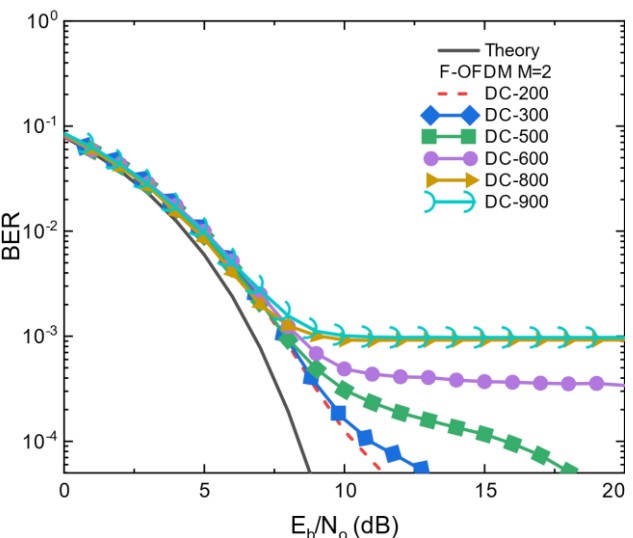

**Figure 11.** BER performance as a function of $E_b/N_o$ for F-OFDM with 2-PAM and 600 AC.

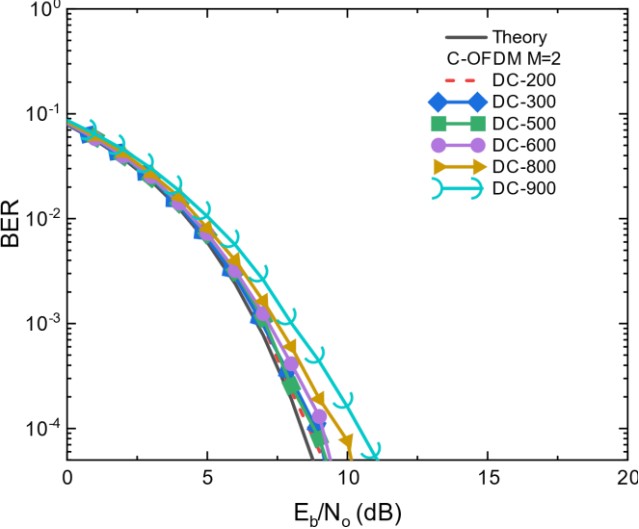

**Figure 12.** BER performance as a function of $E_b/N_o$ for C-OFDM with 2-PAM and 600 AC.

In general, in this section, we explained the change in the performance of the BER as a function of $E_b/N_o$ when several bias points changed, along with the DC level. As can be observed in Figures 6, 9 and 12 that the BER target was achieved with the lowest power penalty for $E_b/N_o$ in comparison with Figures 4, 5, 7, 8, 10 and 11. Moreover, a remarkable improvement was achieved by comparing this power penalty between the conventional OFDM, F-OFDM and C-OFDM under the same bias point conditions and DC level displays. All the previous comparisons showed the advantages of C-OFDM over the normal OFDM and F-OFDM.

### 3.2. Power Penalties

Analysing the optical power penalty for the signals under test, as depicted in Figure 13, provided additional insight for the C-OFDM and F-OFDM using 2-PAM for the $10^{-3}$ BER target. As a result, the power penalty (in dB) increased exponentially with the increase in the DC bias points compared to the theoretical BER curve. Additionally, there was an ~2.5 dB difference between the two modulation techniques when the LED's nonlinear effect was considered.

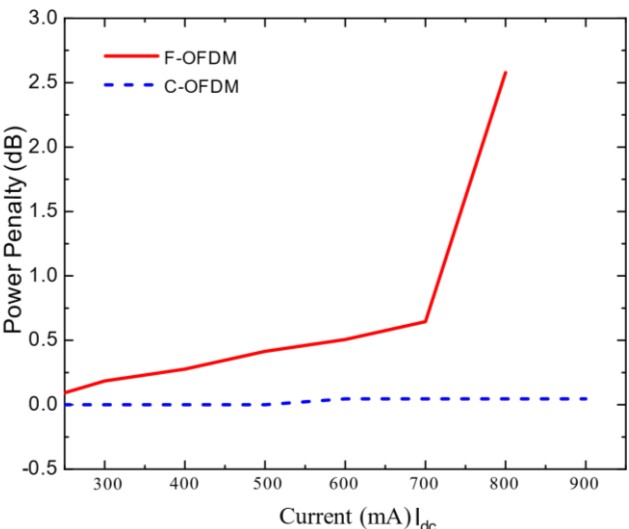

**Figure 13.** The net power penalty gains for C-OFDM over F-OFDM at 200 mA.

Finally, Figure 14 illustrates the PAPR measured for the C-OFDM, F-OFDM and conventional OFDM. The PAPR was given as follows:

$$PAPR = 10 \log 10 \left[ \frac{max|x_m|^2}{E\left[|x_m|^2\right]} \right] \tag{5}$$

where $E[.]$ denotes the average power of the data and $x_m$ is the OFDM signal.

Simulation results showed that the C-OFDM, FOFDM and OFDM systems indicated that C-OFDM achieved the lowest PAPR for values of N of 16 and 64. Reductions of $\cong$2 dB and 5 dB PAPR, respectively, were performed relative to F-OFDM and OFDM. That was because DBS reduced the likelihood of the constructive superposition of the input samples.

Moreover, it was apparent that the PAPR of C-OFDM was the lowest compared to the F-OFDM and OFDM. The reason why the conventional OFDM had a higher PAPR for any value of *N* was due to the cosine and sine components. However, in C-OFDM and F-OFDM, only the cosine component contributed to the power signal. Moreover, concerning the C-transformer, more than 65% of its elements were zero, which reduced the probability of obtaining a high PAPR compared to the F-OFDM. Additionally, the simulation results above presented two significant improvements concerning C-OFDM over F-OFDM and the conventional OFDM. Firstly, C-OFDM demonstrated an enhanced performance under the same signal conditions. Secondly, it obtained a lower power penalty and PAPR. Due to

the limitations imposed by non-linearity transmission devices, a lower power penalty and PAPR were advantageous for communication systems that were achieved with C-OFDM.

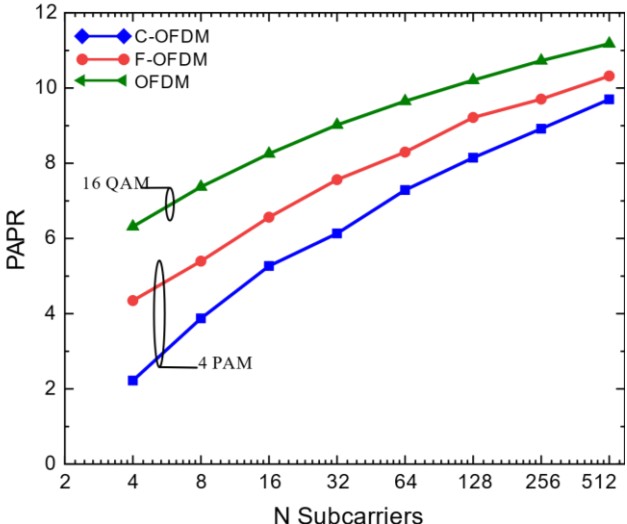

**Figure 14.** The measured PAPR for C-OFDM, F-OFDM and OFDM.

## 4. Conclusions

The LED non-linearity phenomenon significantly impacted the performance of the optical OFDM system. The DC bias point was one of the key characteristics that influenced the distortion in the LED-transmitted information. In this paper, we proposed C-OFDM to investigate the influence of the LED nonlinear behaviour in optical OFDM wireless communication systems. We comparatively simulated the BER performance of C-OFDM against F-OFDM and the conventional OFDM. The results confirmed that a BER of $10^{-4}$ could be achieved with C-OFDM at the 900 mA bias point, and that it could not be achieved with the aforementioned OFDM systems. Specifically, the linear region was extended. However, when F-OFDM was employed, we needed to increase the power by 2.5 to realise the BER at 800 mA compared to the C-OFDM. Consequently, the proposed system delivered a better linear region in comparison to F-OFDM and the traditional OFDM. Finally, the C-OFDM system needed a lower energy-per-bit to noise spectral density ratio ($E_b/N_o$) than standard systems, according to simulation data. As a result, across LED nonlinear regions, a multi-carrier system using the proposed technique would benefit from a low PAPR, and had lower $E_b/N_o$ needs.

**Author Contributions:** Conceptualization, J.A. and S.B.; methodology, S.B.; software, J.A.; writing—original draft preparation, J.A.; writing—review and editing, S.B. All authors have read and agreed to the published version of the manuscript.

**Funding:** This research received no external funding.

**Institutional Review Board Statement:** Not applicable.

**Informed Consent Statement:** Not applicable.

**Data Availability Statement:** The data presented in this study are available on request from the corresponding author.

**Conflicts of Interest:** The authors declare no conflict of interest.

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
