# Peer review of "Visible Light Communication: An Investigation of LED Non-Linearity Effects on VLC Utilising C-OFDM"

_photonics, doi:10.3390/photonics9030192_

Round 1

Reviewer 1 Report

Manuscript can be accepted 

Author Response

Dear Reviewers,
We appreciate you and the reviewers taking the time to read our work and provide insightful feedback. Your informative and helpful remarks influenced future changes in the current version. The writers attentively studied the comments and made every effort to answer each one. We trust that the text, after many modifications, meets your exacting requirements. The authors would appreciate any more helpful remarks.

Regards

Jummah

Reviewer 2 Report

The paper needs effort to be read. This reviewer has several concerns. Besides, I think the topic is quite far away from the focus of the journal, I would suggest to send to a different journal.

The first one starts with the title. The title is erroneous. Looking at this title, you would expect to find what is exposed in the tile, but the only thing you find is a some simulations on C-OFDM and its BER.

At the introduction, most of the aspect are missing. There are many other techniques. The state of the art is incomplete. The state of the art at the introduction is a little bit mess. It should be completely rewritten, oragnized and including the other methods such as predistortion, ACE-like, coding...

The part when introducing the techniques for VLC such as ACO and DCO must be clearly introduced and explained.

In section 2. Authors should include some comments on the penlaty (if any) on efficiency due to the real signals requirements for C-OFDM. Please provide references to support your discussion.

I think I missed something. In eq at line 109 and 110, Sch(7) does not means the sign has changed 7 times, always changes 4 times except the first one. So the question is, what Sch(7) means?. It is seems that is an index, but index of what?

The matemathical formualtion and discussion is very hard to follow because there are many undefined values, or they are unclear or messed up.

About results:

The authors should expliain the LED nonlinearity model and parameters, that would definitively affect the performance.

How eb/N0 has been calculated for the different systems? It must be clearly stated. We are dealing with real signals and complex signals. We are dealing with N/2 loaded subcarriers, or N/4 loaded subcarriers, or N loaded subcarriers. This is very important for a fair comparison, being N the number of subcarriers.

Since the figures are drawn with a single parameter and they are bad explained, it is very hard to conclude with the same arguments used by the authors.

A comparison with other techniques for PAPR reduction in VLC must be provided.

About figures:

I think the figures should be re done.

Fig. 3 for example. Figure unclear. What that means 200_100 or 200_300

It seams to be related to the bias point 200 mA

It also seams to be related to I AC and I DC, but it should be better explained. Moreover, if the cursve is always with I AC = 200 mA, why to remove that and put into the caption?

I would suggest to combine several figures removing some extra information for comparison pruposes. Besides, I would suggest to use markers because if you print the document in B&W you will see nothing.

You should make a figure to compare the different architectures or systems with the same parameters to be able to see if there are benefits or not in their proposal.

Figure 12: With respect whihc system is being compared. I mean, 0 dB is a comparison of what?

Figure 13: This figure is the only one where you can see something in comparison. however, it is not explained how it has been obtained. More explanation on this should be provided.

Minor issues:

Please use DC and AC in capitals instead of in lower face.

bias? In line 176 I guess that bise means bias.

Round 2

Reviewer 2 Report

The authors have accomplished few of my suggestions/recommendations/changes. 

the explanations are still unclear and they need to be better explained and included in the paper, not only in the response to reviewers. 

The state of the art should be completed. A fair comparison is mandatory. 

Besides, I think I missed something, because the explanation on how many subcarriers are being used is a little bit confuse. Do you mean that you are using a multicarrier system but you only use PAM modulations instead of QAM, PSK modulations to avoid the reduction of half of the subcarriers? Even with PAM modulation is a multicarrier you need to use the hermitic to overcome this effect.

I think the authors should continue answering the Reviewers' suggestions in a better way for the clarity of the paper for the readers and fairness in comparison.

Author Response

Dear reviewer, please accept my heartfelt greetings. 

We would appreciate your precious time reviewing our paper and providing valuable comments. Your valuable and insightful comments led to possible improvements in the current version. The authors have carefully considered the statements and tried their best to address every one of them. We hope the manuscript, after careful revisions, meet your high standards. The authors welcome further constructive comments, if any.

Could you please check the attachment as a response to your concerns and remarks.

Sincerely,

Authors 
